# Optimistic Meta-Gradients

**Sebastian Flennerhag**
Google DeepMind
flennerhag@google.com

**Tom Zahavy**
Google DeepMind

**Brendan O'Donoghue**
Google DeepMind

**Hado van Hasselt**
Google DeepMind

**András György**
Google DeepMind

**Satinder Singh**
Google DeepMind

## Abstract

We study the connection between gradient-based meta-learning and convex optimisation. We observe that gradient descent with momentum is a special case of meta-gradients, and building on recent results in optimisation, we prove convergence rates for meta-learning in the single task setting. While a meta-learned update rule can yield faster convergence up to constant factor, it is not sufficient for acceleration. Instead, some form of optimism is required. We show that optimism in meta-learning can be captured through the recently proposed Bootstrapped Meta-Gradient [9] method, providing deeper insight into its underlying mechanics.

## 1 Introduction

In meta-learning, a learner is using a parameterised algorithm to adapt to a given task. The parameters of the algorithm are then meta-learned by evaluating the learner's resulting performance [25, 10, 2]. As such, meta-learning features a complex interaction between the learner and the meta-learner. The **learner's problem** is to minimize the expected loss $f$ of a stochastic objective by adapting its parameters $x \in \mathbb{R}^n$. The learner has an update rule $\varphi$ at its disposal that generates new parameters $x_t = x_{t-1} + \varphi(x_{t-1}, w_t)$; we suppress data dependence to simplify notation. A simple example is when $\varphi$ represents gradient descent with $w_t = \eta$ its step size, that is $\varphi(x_{t-1}, \eta) = -\eta \nabla f(x_{t-1})$ [17, 26]; several works have explored meta-learning other aspects of a gradient-based update rule [6, 21, 7, 30, 31, 9, 15, 22]. $\varphi$ need not be limited to a gradient-based update, it can represent some algorithm implemented within a Neural Network [25, 11, 1, 29].

**The meta-learner's problem** is to optimise the meta-parameters $w_t$ to yield effective updates.

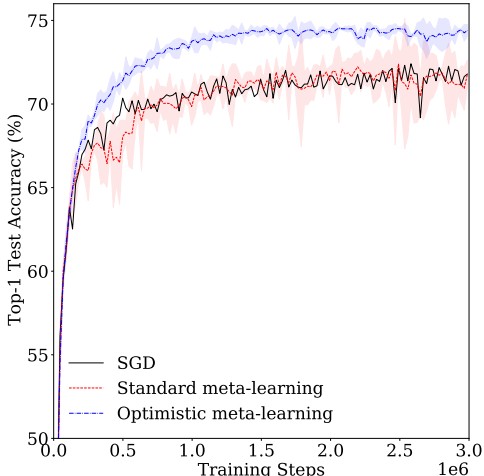

Figure 1: ImageNet. We compare training a 50-layer ResNet using SGD against variants that tune an element-wise learning rate online using standard meta-learning or optimistic meta-learning. Shading depicts 95% confidence intervals over 3 seeds.

In a typical (gradient-based) meta-learning setting, it does so by treating $x_t$ as a function of $w$. Let $h_t$, defined by $h_t(w) = f(x_{t-1} + \varphi(x_{t-1}, w))$, denote the learner's post-update performance as a function of $w$. The learner and the meta-learner co-evolve according to

$$x_t = x_{t-1} + \varphi(x_{t-1}, w_t), \qquad w_{t+1} = w_t - \nabla h_t(w_t) = w_t - D\varphi(x_{t-1}, w_t)^T \nabla f(x_t),$$

37th Conference on Neural Information Processing Systems (NeurIPS 2023).

where $D\varphi(x, w)$ denotes the Jacobian of $\varphi$ with respect to $w$. The nested structure between these two updates makes it challenging to analyse meta-learning, in particular it depends heavily on the properties of the Jacobian. In practice, $\varphi$ is highly complex and so $D\varphi$ is almost always intractable. For this reason, the only theoretical results we are aware of specialise to the multi-task setting, where the learner must adapt to a new task $f_t$. Acceleration in this setup is driven by the tasks similarity. That is, if all tasks are sufficiently similar, a meta-learned update can accelerate convergence [14]. However, these results do not yield acceleration in the absence of a task distribution to the best of our knowledge.

This paper provides an alternative view. We study the classical convex optimisation setting of approximating the minimiser $\min_x f(x)$. We observe that setting the update rule equal to the gradient, i.e. $\varphi : (x, w) \mapsto w\nabla f(x)$, recovers gradient descent. Similarly, we show in Section 4 that $\varphi$ can be chosen to recover gradient descent with momentum. This offers another view of meta-learning as a non-linear transformation of classical optimisation. An implication thereof is that task similarity is not necessary condition for improving the rate of convergence via meta-learning. While there is ample empirical evidence to that effect [30, 31, 9, 16], we are only aware of theoretical results in the special case of meta-learned step sizes [17, 26].

Given a function $f$ that is convex with Lipschitz smooth gradients, meta-learning improves the rate of convergence by a multiplicative factor of $\lambda$ to $O(\lambda/T)$ via the smoothness of the update rule. To achieve accelerated convergence, $O(1/T^2)$, some form of *optimism* is required, typically in the form of a prediction of the next gradient. We consider optimism with meta-learning in the convex setting and prove accelerated rates of convergence, $O(\lambda/T^2)$. Again, meta-learning affects these bounds by a multiplicative factor. Our main contributions are as follows:

1. We show that meta-gradients contain gradient descent with momentum (Heavy Ball [23]; Section 4) and Nesterov Acceleration [20] as special cases (Section 5).
2. We show that gradient-based meta-learning can be understood as a non-linear transformation of an underlying optimisation method (Section 4).
3. We establish rates of convergence for meta-learning in the convex setting (Section 4).
4. We show that optimism can be expressed through the recently proposed Bootstrapped Meta-Gradient method [BMG; 9]. Our analysis provides a first proof of convergence for BMG and highlights the underlying mechanics that enable faster learning with BMG (Section 6).

## 2 Meta-Learning as Convex Optimisation

**Problem definition.** This section defines the problem studied in this paper and introduces our notation (see Appendix A, Table 1). Let $f : \mathcal{X} \to \mathbb{R}$ be a proper and convex function. The problem of interest is to approximate the global minimum $\min_{x \in \mathcal{X}} f(x)$. We assume a global minimiser exists and is unique, defined by

$$x^* = \arg\min_{x \in \mathcal{X}} f(x). \tag{1}$$

We assume that $\mathcal{X} \subseteq \mathbb{R}^n$ is a closed, convex and non-empty set. $f$ is differentiable and has Lipschitz smooth gradients with respect to a norm $\|\cdot\|$, meaning that there exists $L \in (0, \infty)$ such that $\|\nabla f(x) - \nabla f(y)\|_* \le L\|x - y\|$ for all $x, y \in \mathcal{X}$, where $\|\cdot\|_*$ is the dual norm of $\|\cdot\|$. We consider the noiseless setting for simplicity; our results carry over to the stochastic setting by replacing the key online-to-batch bound used in our analysis by its stochastic counterpart [13].

**Algorithm.** Let $[T] = \{1, 2, \ldots, T\}$. We are given weights $\{\alpha_t\}_{t=1}^T$, each $\alpha_t > 0$, and an initialisation $(\bar{x}_0, w_1) \in \mathcal{X} \times \mathcal{W}$. At each time $t \in [T]$, an update rule $\varphi : \mathcal{X} \times \mathcal{W} \to \mathcal{X}$ generates the update $x_t = \varphi(\bar{x}_{t-1}, w_t)$, where $\mathcal{W} \subseteq \mathbb{R}^m$ is closed, convex, and non-empty. We discuss $\varphi$ momentarily. The algorithm maintains the online average

$$\bar{x}_t = \frac{x_{1:t}}{\alpha_{1:t}} = (1 - \rho_t)\bar{x}_{t-1} + \rho_t x_t, \tag{2}$$

where $x_{1:t} = \sum_{s=1}^t \alpha_s x_s$, $\alpha_{1:t} = \sum_{s=1}^t \alpha_s$, and $\rho_t = \alpha_t/\alpha_{1:t}$. Our goal is to establish conditions under which $\{\bar{x}_t\}_{t=1}^T$ converges to the minimiser $x^*$. While this moving average is not always used in practical applications, it is required for accelerated rates in online-to-batch conversion [27, 3, 13].

| **Algorithm 1:** Meta-learning in practice. | **Algorithm 2:** Meta-learning in the convex setting. |
|---|---|
| **input :** Weights $\{\beta_t\}_{t=1}^T$ 
 **input :** Update rule $\varphi$ 
 **input :** Initialisation $(x_0, w_1)$ 
 **for** $t = 1, 2, \ldots, T$: 
 $\quad x_t = x_{t-1} + \varphi(x_{t-1}, w_t)$ 
 $\quad h_t(\cdot) = f(x_{t-1} + \rho_t \varphi(x_{t-1}, \cdot))$ 
 $\quad w_{t+1} = w_t - \beta_t \nabla h_t(w_t)$ 
 **return** $x_T$ | **input :** Weights $\{\alpha_t\}_{t=1}^T, \{\beta_t\}_{t=1}^T$ 
 **input :** Update rule $\varphi$ 
 **input :** Initialisation $(\bar{x}_0, w_1)$ 
 **for** $t = 1, 2, \ldots, T$: 
 $\quad x_t = \varphi(\bar{x}_{t-1}, w_t)$ 
 $\quad \bar{x}_t = (1 - \alpha_t/\alpha_{1:t})\bar{x}_{t-1} + (\alpha_t/\alpha_{1:t})x_t$ 
 $\quad g_t = D\varphi(\bar{x}_{t-1}, w_t)^T \nabla f(\bar{x}_t)$ 
 $\quad w_{t+1} = \arg\min_{w \in \mathcal{W}} \sum_{s=1}^t \alpha_s \langle g_s, w \rangle + \frac{1}{2\beta_t}\|w\|^2$ 
 **return** $\bar{x}_T$ |

Convergence depends on how meta-parameters $w_t$ are chosen. The meta-learner faces a sequence of losses $h_t : \mathcal{W} \to \mathbb{R}$ defined by the composition $h_t(w) = f((1 - \rho_t)\bar{x}_{t-1} + \rho_t\varphi(\bar{x}_{t-1}, w))$. As such, the meta-learner is facing an online optimization, which we model under Follow-The-Regularized-Leader (FTRL; reviewed in Section 3): given $w_0$, each $w_t$ is chosen according to

$$w_{t+1} = \underset{w \in \mathcal{W}}{\arg\min} \left( \sum_{s=1}^t \alpha_s \langle \nabla h_s(w_s), w \rangle + \frac{1}{2\beta}\|w\|^2 \right). \tag{3}$$

Note that Eq. 3 subsumes the standard meta-gradient; if $\|\cdot\|$ is the Euclidean norm, an interior solution yields $w_{t+1} = w_t - \alpha_t\beta\nabla h_t(w_t)$. It is straightforward to extend Eq. 3 to account for meta-updates that use AdaGrad-like [5] acceleration by altering the norms (see [12]).

**Update rule.** It is not possible to prove convergence outside of the convex setting, since $\varphi$ may reach a local minimum, where local changes to $w$ do not yield better updates in $x$, yet the $x$ sequence is not converging. Convexity means that each $h_t$ must be convex, which requires that $\varphi$ is affine in $w$ (but may vary non-linearly in $x$). We also assume that $\varphi$ is smooth with respect to $\|\cdot\|$, in the sense that it has bounded norm; for all $x \in \mathcal{X}$ and all $w \in \mathcal{W}$ we assume that there exists $\lambda \in (0, \infty)$ for which

$$\|D\varphi(x, w)^T \nabla f(x)\|_*^2 \leq \lambda \|\nabla f(x)\|_*^2.$$

**Limitations.** Our analysis makes relatively strict assumptions. Most meta-learning systems are not affine in $w$. A notable case where our assumptions to hold is meta-learned step-sizes or preconditioning matrices. For other update rules, our analysis holds up to first-order Taylor approximation error. We carry out experiments in Section 7 to empirically verify our theoretical insights.

## 3 Preliminaries: Online Convex Optimisation

In this section, we present analytical tools from the optimisation literature that we build upon. In a standard optimisation setting, there is no update rule $\varphi$; instead, the iterates $x_t$ are generated by a gradient-based algorithm, akin to Eq. 3. Our problem setting reduces to standard optimisation if $\varphi$ is defined by $\varphi : (x, w) \mapsto w$, in which case $x_t = w_t$. In this paper, we use batch-to-online conversion as our analytical tool. This strategy treats the iterates $x_1, x_2, \ldots$ as generated by an online learning algorithm, for which we can obtain a regret bound. This regret bound can then be turned into a convergence rate, detailed momentarily.

**Online Optimisation.** In online convex optimisation [32], a learner is given a convex decision set $\mathcal{U}$ and faces a sequence of convex loss functions $\{\alpha_t f_t\}_{t=1}^T$. At each time $t \in [T]$, it must make a prediction $u_t$ prior to observing $\alpha_t f_t$, after which it incurs a loss $\alpha_t f_t(u_t)$ and receives a signal—either $\alpha_t f_t$ itself or a (sub-)gradient of $\alpha_t f_t(u_t)$. The learner's goal is to minimise *regret*, $R(T) := \sum_{t=1}^T \alpha_t(f_t(u_t) - f_t(u))$, against a comparator $u \in \mathcal{U}$. An important property of a convex function $f$ is $f(u') - f(u) \leq \langle \nabla f(u'), u' - u \rangle$. Hence, the regret is largest under linear losses: $\sum_{t=1}^T \alpha_t(f_t(u_t) - f_t(u)) \leq \sum_{t=1}^T \alpha_t \langle \nabla f_t(u_t), u_t - u \rangle$. For this reason, it is sufficient to consider regret under linear loss functions. An algorithm has sublinear regret if $\lim_{T \to \infty} R(T)/T = 0$.

**FTRL & AO-FTRL.** The meta-update in Eq. 3 is an instance of Follow-The-Regularised-Leader (FTRL) under linear losses. In Section 6, we show that BMG is an instance of the Adaptive-Optimistic

FTRL (AO-FTRL) [24, 19, 13, 28]. In AO-FTRL, we have a strongly convex regulariser $\|\cdot\|^2$. AO-FTRL sets the first prediction $u_1$ to minimise $\|\cdot\|^2$. Given linear losses $\{g_s\}_{s=1}^{t-1}$ and learning rates $\{\beta_t\}_{t=1}^T$, each $\beta_t > 0$, the algorithm proceeds according to

$$u_t = \arg\min_{u\in\mathcal{U}} \left( \alpha_t\langle\tilde{g}_t, u\rangle + \sum_{s=1}^{t-1}\alpha_s\langle g_s, u\rangle + \frac{1}{2\beta_t}\|u\|^2 \right), \tag{4}$$

where each $\tilde{g}_t$ is a "hint" that enables optimistic learning [24, 19]; setting $\tilde{g}_t = 0$ recovers the original FTRL algorithm. The goal of a hint is to predict the next loss vector $g_t$; if the predictions are accurate AO-FTRL can achieve lower regret than its non-optimistic counter-part. Since $\|\cdot\|^2$ is strongly convex, FTRL is well defined in the sense that the minimiser exists, is unique, and finite [18]. The regret of FTRL and AO-FTRL against any comparator $u \in \mathcal{U}$ can be upper-bounded by

$$R(T) = \sum_{t=1}^T \alpha_t\langle g_t, u_t - u\rangle \leq \frac{\|u\|^2}{2\beta_T} + \frac{1}{2}\sum_{t=1}^T \alpha_t^2\beta_t\|g_t - \tilde{g}_t\|_*^2. \tag{5}$$

Hence, hints that predict $g_t$ well can reduce the regret substantially. Without hints, FTRL can guarantee $O(\sqrt{T})$ regret (for non strongly convex loss functions). However, [4] show that under linear losses, if hints are weakly positively correlated—defined as $\langle g_t, \tilde{g}_t\rangle \geq \epsilon\|g_t\|^2$ for some $\epsilon > 0$—then the regret guarantee improves to $O(\log T)$, even for non strongly-convex loss functions. We believe optimism provides an exciting opportunity for novel forms of meta-learning. Finally, we note that these regret bounds (and hence our analysis) can be extended to stochastic optimisation [19, 12].

**Online-to-batch conversion.** The main idea behind online to batch conversion is that, for $f$ convex, Jensen's inequality gives $f(\bar{x}_T) - f(x^*) \leq \sum_{t=1}^T \alpha_t\langle\nabla f(x_t), x_t - x^*\rangle/\alpha_{1:T}$. That is, the sub-optimality gap of the average iterate $\bar{x}_T$ can be bounded by the regret in the sequence $x_1, x_2, \ldots, x_T$. Applying this bound naively yields $O(1/T)$ rate of convergence. In recent work, [3] shows that one achieve tighter bounds by instead querying the gradient at the *average* iterate, $f(\bar{x}_T) - f(x^*) \leq \sum_{t=1}^T \alpha_t\langle\nabla f(\bar{x}_t), x_t - x^*\rangle/\alpha_{1:T}$. A tighter bound means a faster rate of convergence. Recently, [13] tightened the analysis further and proved that the sub-optimality gap can be bounded by

$$f(\bar{x}_T) - f(x^*) \leq$$
$$\frac{1}{\alpha_{1:T}} \left( R^x(T) - \frac{\alpha_t}{2L}\|\nabla f(\bar{x}_t) - \nabla f(x^*)\|_*^2 - \frac{\alpha_{1:t-1}}{2L}\|\nabla f(\bar{x}_{t-1}) - \nabla f(\bar{x}_t)\|_*^2 \right), \tag{6}$$

were we define $R^x(T) := \sum_{t=1}^T \alpha_t\langle\nabla f(\bar{x}_t), x_t - x^*\rangle$ as the regret of the sequence $\{x_t\}_{t=1}^T$ against the comparator $x^*$. With this machinery in place, we now turn to deriving our main results.

## 4  Meta-Gradients without Optimism

The main difference between classical optimisation and meta-learning is the introduction of the update rule $\varphi$. To see how this acts on optimisation, consider two special cases. If the update rule just return the gradient, $\varphi = \nabla f$, Eq. 3 reduces to gradient descent (with averaging). This inductive bias is fixed and does not change with experience, so acceleration is not possible: the rate of convergence is $O(1/\sqrt{T})$ [28]. The other extreme is an update rule that only depends on the meta-parameters, $\varphi(x, w) = w$. Here, the meta-learner has ultimate control and selects the next update without constraints. The only relevant inductive bias is contained in $w$. To see how this inductive bias is formed, suppose $\|\cdot\| = \|\cdot\|_2$ so that Eq. 3 yields $w_{t+1} = w_t - \alpha_t\rho_t\beta\nabla f(\bar{x}_t)$ (assuming an interior solution). Combining this with the moving average in Eq. 2, we may write the learner's iterates as

$$\bar{x}_t = \bar{x}_{t-1} + \tilde{\rho}_t(\bar{x}_{t-1} - \bar{x}_{t-2}) - \tilde{\beta}_t\nabla f(\bar{x}_{t-1}),$$

where each $\tilde{\rho}_t = \rho_t\frac{1-\rho_{t-1}}{\rho_{t-1}}$ and $\tilde{\beta}_t = \alpha_t\rho_t\beta$. Setting $\beta = 1/(2L)$ and each $\alpha_t = t$ yields $\tilde{\rho}_t = \frac{t-2}{t+1}$ and $\tilde{\beta}_t = t/(4(t+1)L)$, which recovers Polyak's canonical Heavy-Ball method [23]. Hence, gradient descent with momentum is a special case of meta-learning under the update rule $\varphi : (x, w) \mapsto w$. Because Heavy Ball carries momentum from past updates, it can encode a model of the learning dynamics that leads to faster convergence, on the order $O(1/T)$. The implication of this is that the dynamics of meta-learning are fundamentally momentum-based and thus learns an inductive bias in the same cumulative manner. This similarity carries in our theoretical analysis, which we turn to next.

The central challenge in applying the bound in Eq. 6 to Algorithm 2 is that the iterates $x_t$ are generated under the update rule $\varphi$. Hence, we cannot apply standard regret bounds directly. Instead, observe that

$$R^x(T) = \sum_{t=1}^{T} \alpha_t \langle \nabla f(\bar{x}_t), x_t - x^* \rangle = \sum_{t=1}^{T} \alpha_t \langle \nabla f(\bar{x}_t), \varphi(\bar{x}_{t-1}, w_t) - x^* \rangle$$

$$= \underbrace{\sum_{t=1}^{T} \alpha_t \langle \nabla f(\bar{x}_t), \varphi(\bar{x}_{t-1}, w_t) - \varphi(\bar{x}_{t-1}, w^*) \rangle}_{\text{regret under losses } \ell_t(\cdot) = \alpha_t \langle \nabla f(\bar{x}_t), \varphi(\bar{x}_{t-1}, \cdot) \rangle} + \underbrace{\sum_{t=1}^{T} \alpha_t \langle \nabla f(\bar{x}_t), \varphi(\bar{x}_{t-1}, w^*) - x^* \rangle}_{\text{difference in comparator capacity}}.$$

The first term in the final inequality can be understood as the regret under convex losses $\ell_t(\cdot) = \alpha_t \langle \nabla f(\bar{x}_t), \varphi(\bar{x}_{t-1}, \cdot) \rangle$. Since $\varphi$ is affine, $\ell_t$ is convex and thus this regret can be upper-bounded by linearising the losses. The linearisation reads $\langle D\varphi(\bar{x}_{t-1}, w_t)^T \nabla f(\bar{x}_t), \cdot \rangle$, which is identical the linear losses $\langle \nabla h_t(w_t), \cdot \rangle$ faced by the meta-learner in Eq. 3. In other words, the regret component can be upper-bounded by the of the meta-learner,

$$R^w(T) := \sum_{t=1}^{T} \alpha_t \langle \nabla h_t(w_t), w_t - w^* \rangle \geq \sum_{t=1}^{T} \alpha_t \langle \nabla f(\bar{x}_t), \varphi(\bar{x}_{t-1}, w_t) - \varphi(\bar{x}_{t-1}, w^*) \rangle.$$

The regret of the learner is therefore upper bounded by

$$R^x(T) \leq R^w(T) + \sum_{t=1}^{T} \alpha_t \langle \nabla f(\bar{x}_t), \varphi(\bar{x}_{t-1}, w^*) - x^* \rangle. \tag{7}$$

The last term captures the difference in comparator capacity and more specifically the amount of regret they can inflict. If the comparator $w^*$ has more power than that of $x^*$, it can accumulate a lower total loss, in which case this term will be negative, allowing us to discard it. Intuitively, the comparator $x^*$ is non-adaptive. It must make one choice $x^*$ and suffer the average loss. In contrast, the comparator $w^*$ becomes adaptive under the update rule; it can only choose one $w^*$, but on each round it plays $\varphi(\bar{x}_{t-1}, w^*)$. If $\varphi$ is sufficiently flexible, this gives the comparator $w^*$ more power than $x^*$, and hence it can force the meta-learner to suffer greater regret. When this is the case, we say that regret is retained when moving from $x^*$ to $w^*$. As long as $\varphi$ is not degenerate, this is typically easy to satisfy by making $\mathcal{W}$ sufficiently large.

**Definition 1.** *Given $f$, $\{\alpha_t\}_{t=1}^{T}$, and $\{x_t\}_{t=1}^{T}$, an update rule $\varphi : \mathcal{X} \times \mathcal{W} \to \mathcal{X}$ preserves regret if there exists a comparator $w \in \mathcal{W}$ that satisfies*

$$\sum_{t=1}^{T} \alpha_t \langle \varphi(\bar{x}_{t-1}, w), \nabla f(\bar{x}_t) \rangle \leq \sum_{t=1}^{T} \alpha_t \langle x^*, \nabla f(\bar{x}_t) \rangle. \tag{8}$$

*If such $w$ exists, let $w^*$ denote the comparator with smallest norm $\|w\|$.*

**Lemma 1.** *Given $f$, $\{\alpha_t\}_{t=1}^{T}$, and $\{x_t\}_{t=1}^{T}$, if $\varphi$ preserves regret, then*

$$R^x(T) = \sum_{t=1}^{T} \alpha_t \langle \nabla f(\bar{x}_t), x_t - x^* \rangle \leq \sum_{t=1}^{T} \alpha_t \langle \nabla f(\bar{x}_t), \varphi(\bar{x}_{t-1}, w_t) - \varphi(\bar{x}_{t-1}, w^*) \rangle = R^w(T).$$

Proof: Appendix D. From Eq. 8, it is clear that for $\varphi$ to retain regret, it must admit a parameterisation that correlates negatively with the gradient. In other words, $\varphi$ must be able to behave as a gradient descent algorithm. However, this must not hold on every step, only sufficiently often. For instance, $\varphi(x, \cdot)$ affine can be made to satisfy this condition if $\mathcal{X}$ and $\mathcal{W}$ are chosen appropriately.

**Theorem 1.** *Let $\varphi$ preserve regret and satisfy the assumptions in Section 2. Then*

$$f(\bar{x}_T) - f(x^*) \leq \frac{1}{\alpha_{1:T}} \left( \frac{\|w^*\|^2}{\beta} + \sum_{t=1}^{T} \frac{\lambda \beta \alpha_t^2}{2} \|\nabla f(\bar{x}_t)\|_*^2 \right.$$

$$\left. - \frac{\alpha_t}{2L} \|\nabla f(\bar{x}_t) - \nabla f(x^*)\|_*^2 - \frac{\alpha_{1:t-1}}{2L} \|\nabla f(\bar{x}_{t-1}) - \nabla f(\bar{x}_t)\|_*^2 \right).$$

*If $x^*$ is a global minimiser of $f$, setting $\alpha_t = 1$ and $\beta = \frac{1}{\lambda L}$ yields $f(\bar{x}_T) - f(x^*) \leq \frac{\lambda L \operatorname{diam}(\mathcal{W})}{T}$.*

| **Algorithm 3:** BMG in practice. | **Algorithm 4:** Convex optimistic meta-learning. |
|---|---|
| **input:** Weights $\{\beta_t\}_{t=1}^T$ | **input:** Weights $\{\alpha_t\}_{t=1}^T, \{\beta_t\}_{t=1}^T$ |
| **input:** Update rule $\varphi$, divergence $B^\mu$ | **input:** Update rule $\varphi$ |
| **input:** Target oracle | **input:** Hints $\{\tilde{g}_t\}_{t=1}^T$ |
| **input:** Initialisation $(x_0, w_1)$ | **input:** Initialisation $(\bar{x}_0, w_1)$ |
| **for** $t = 1, 2, \ldots, T$: | **for** $t = 1, 2, \ldots, T$: |
| $\quad x_t = x_{t-1} + \varphi(x_{t-1}, w_t)$ | $\quad x_t = \varphi(\bar{x}_{t-1}, w_t)$ |
| $\quad$ Query $z_t$ from target oracle | $\quad \bar{x}_t = (1 - \alpha_t/\alpha_{1:t})\bar{x}_{t-1} + (\alpha_t/\alpha_{1:t})x_t$ |
| $\quad d_t(\cdot) = B^\mu_{z_t}(x_{t-1} + \varphi(x_{t-1}, \cdot))$ | $\quad g_t = D\varphi(\bar{x}_{t-1}, w_t)^T \nabla f(\bar{x}_t)$ |
| $\quad w_{t+1} = w_t - \beta_t \nabla d_t(w_t)$ | $\quad v_t = \alpha_{t+1}\tilde{g}_{t+1} + \sum_{s=1}^t \alpha_s g_s$ |
| **return** $x_T$ | $\quad w_{t+1} = \arg\min_{w \in \mathcal{W}} \langle v_t, w \rangle + \frac{1}{2\beta_t}\|w\|^2$ |
| | **return** $\bar{x}_T$ |

Proof: Appendix D. Compared to Heavy Ball, meta-learning introduces a constant $\lambda$ that captures the smoothness of the update rule. Hence, while meta-learning does not achieve better scaling in $T$ through $\varphi$, it can improve upon classical optimisation by a constant factor if $\lambda < 1$.

That meta-learning can improve upon momentum is borne out experimentally. We consider the problem of minimizing an ill-conditioned convex quadratic and compare standard momentum to a version with meta-learned step-size, i.e. $\varphi : (x, w) \mapsto w \odot \nabla f(x)$, where $\odot$ is the Hadamard product. We find that introducing a non-linearity $\varphi$ leads to a sizeable improvement in the rate of convergence. See Section 7.1 for further details.

## 5 Meta-Gradients with Optimism

It is well known that minimizing a smooth convex function admits convergence rates of $O(1/T^2)$. Our analysis of meta-learning does not achieve these rates. Previous work indicate that we should not expect it to either; to achieve the theoretical lower-limit of $O(1/T^2)$, some form of *optimism* (c.f. Section 3) is required. A typical form of optimism is to predict the next gradient. This is how Nesterov Acceleration operates [20], and is the reason for its $O(1/T^2)$ convergence guarantee.

From our perspective, meta-learning is a non-linear transformation of the iterate $x$. Hence, we should expect optimism to play a similarly crucial role. Formally, optimism comes in the form of *hint functions* $\{\tilde{g}_t\}_{t=1}^T$, each $\tilde{g}_t \in \mathbb{R}^m$, that are revealed to the meta-learner prior to selecting $w_{t+1}$. These hints give rise to *Optimistic Meta-Learning* (OML) via meta-updates

$$w_{t+1} = \arg\min_{w \in \mathcal{W}} \left( \alpha_{t+1}\tilde{g}_{t+1} + \sum_{s=1}^t \alpha_s \langle \nabla h_s(w_s), w \rangle + \frac{1}{2\beta_t}\|w\|^2 \right). \quad (9)$$

If the hints are accurate, meta-learning with optimism can achieve an accelerated rate of $O(\tilde{\lambda}/T^2)$, where $\tilde{\lambda}$ is a constant that characterises the smoothness of $\varphi$, akin to $\lambda$. Again, we find that meta-learning behaves as a non-linear transformation of classical optimism and its rate of convergence is governed by the geometry it induces.

For a complete description, see Algorithm 4. These updates do not correspond to the typical meta-update in Algorithm 1; however, we show momentarily that they can be interpreted as the targets in the BMG method, summarised in Algorithm 3. Before turning to BMG, we establish that optimistic meta-learning in the convex setting does indeed yield acceleration.

**Theorem 2.** *Let $\varphi$ preserve regret and assume Algorithm 4 satisfy the assumptions in Section 2. Then*

$$f(\bar{x}_T) - f(x^*) \leq \frac{1}{\alpha_{1:T}} \left( \frac{\|w^*\|^2}{\beta_T} + \sum_{t=1}^T \frac{\alpha_t^2 \beta_t}{2} \|D\varphi(\bar{x}_{t-1}, w_t)^T \nabla f(\bar{x}_t) - \tilde{g}_t\|_*^2 \right.$$

$$\left. - \frac{\alpha_t}{2L}\|\nabla f(\bar{x}_t) - \nabla f(x^*)\|_*^2 - \frac{\alpha_{1:t-1}}{2L}\|\nabla f(\bar{x}_{t-1}) - \nabla f(\bar{x}_t)\|_*^2 \right).$$

Proof; Appendix D. From Theorem 2, it is clear that if $\tilde{g}_t$ is a good predictor of $D\varphi(\bar{x}_{t-1}, w_t)^T \nabla f(\bar{x}_t)$, then the positive term in the summation can be cancelled by the negative term. In a classical optimisation setting, $D\varphi = I_n$, and hence it is easy to see that simply

choosing $\tilde{g}_t$ to be the previous gradient is sufficient to achieve the cancellation [13]. Indeed, this choice gives us Nesterov's Accelerated rate [28]. The upshot of this is that we can specialise Algorithm 4 to capture Nesterov's Accelerated method by choosing $\varphi : (x, w) \mapsto w$—as in the reduction to Heavy Ball—and setting the hints to $\tilde{g}_t = \nabla f(\bar{x}_{t-1})$. Hence, while the standard meta-update without optimism contains Heavy Ball as a special case, the optimistic meta-update contains Nesterov Acceleration as a special case.

In the meta-learning setting, $D\varphi$ is not an identity matrix, and hence the best targets for meta-learning are different. Naively choosing $\tilde{g}_t = D\varphi(\bar{x}_{t-1}, w_t)^T \nabla f(\bar{x}_{t-1})$ would lead to a similar cancellation, but this is not allowed. At iteration $t$, we have not computed $w_t$ when $\tilde{g}_t$ is chosen, and hence $D\varphi(\bar{x}_{t-1}, w_t)$ is not available. The nearest term that is accessible is $D\varphi(\bar{x}_{t-2}, w_{t-1})$.

**Corollary 1.** *Let each $\tilde{g}_{t+1} = D\varphi(\bar{x}_{t-1}, w_t)^T \nabla f(\bar{x}_t)$. Assume that $\varphi$ satisfies*

$$\left\| D\varphi(x', w)^T \nabla f(x) - D\varphi(x'', w')^T \nabla f(x') \right\|_*^2 \leq \tilde{\lambda} \left\| \nabla f(x') - \nabla f(x) \right\|_*^2$$

*for all $x'', x', x \in \mathcal{X}$ and $w, w' \in \mathcal{W}$, for some $\tilde{\lambda} > 0$. If each $\alpha_t = t$ and $\beta_t = \frac{t-1}{2t\tilde{\lambda}L}$, then $f(\bar{x}_T) - f(x^*) \leq \frac{4\tilde{\lambda}L\,\mathrm{diam}(\mathcal{W})}{T^2-1}$.*

Proof: Appendix D.

These predictions hold empirically in a non-convex setting. We train a 50-layer ResNet using either SGD with a fixed learning rate, or an update rule that adapts a per-parameter learning rate online, $\varphi : (x, w) \mapsto w \odot \nabla f(x)$. We compare the standard meta-learning approach without optimism to optimistic meta-learning. Figure 1 shows that optimism is critical for meta-learning to achieve acceleration, as predicted by theory (experiment details in Appendix C).

## 6 Bootstrapped Meta-Gradients as a form of Optimism

Given Theorem 2, it is of interest to study practical ways of implementing optimism in meta-learning. We study a recently proposed variant of meta-gradients, *Bootstrapped Meta-Gradients (BMG)* [8]. Informally, instead of directly minimising the loss $f$, the meta-objective in BMG is the distance between the meta-learner's output $x_t$ and a desired *target* $z_t$. The target is computed by unrolling the meta-learner for a further number of steps, thus implicitly embodying a form of optimism, before a gradient step is taken: $z_t = x_t + \varphi(x_t, w_t) - \nabla f(x_t + \varphi(x_t, w_t))$. This encodes optimism via $\varphi$ because it encourages the meta-learner to build up momentum (i.e. to accumulate past updates). To see this formally, we turn to AO-FTRL. First, we provide a more general definition of BMG. Let $\mu : \mathcal{X} \to \mathbb{R}$ be a convex distance generating function and define the Bregman Divergence $B^\mu : \mathbb{R}^n \times \mathbb{R}^n \to \mathbb{R}$ by

$$B_z^\mu(x) = \mu(x) - \mu(z) - \langle \nabla\mu(z), x - z \rangle.$$

Given initial condition $(x_0, w_1)$, the BMG updates proceed according to

$$x_t = x_{t-1} + \varphi(x_{t-1}, w_t)$$
$$w_{t+1} = w_t - \beta_t \nabla d_t(w_t)$$
$$= w_t - \beta_t D\varphi(x_{t-1}, w_t)^T \left( \nabla\mu(x_t) - \nabla\mu(z_t) \right), \tag{10}$$

where $d_t : \mathbb{R}^n \to \mathbb{R}$ is defined by $d_t(w) = B_{z_t}^\mu(x_{t-1} + \varphi(x_{t-1}, w_t))$; each $z_t \in \mathbb{R}^n$ is referred to as a target. See Algorithm 3 for an algorithmic summary. To show how this relates to AO-FTRL, let $\mu = f$. In this case, the BMG update reads $w_{t+1} = w_t - \beta_t D\varphi(x_{t-1}, w_t)^T (\nabla f(z_t) - \nabla f(x_t))$. We can obtain these updates via our convex framework (i.e. Algorithm 4) by setting $\tilde{g}_{t+1} = \nabla f(z_t)$. In this case, we have that (Corollary 3, Appendix E) AO-FTRL reduces to

$$w_{t+1} = \frac{\beta_{t+1}}{\beta_t} w_t - \beta_t D\varphi(\bar{x}_{t-1}, w_t)^T (\alpha_{t+1}\nabla f(z_t) - \alpha_t \nabla f(\bar{x}_t)) + \xi_t,$$

where $\xi_t = \beta_t \alpha_t D\varphi(\bar{x}_{t-2}, w_{t-1})^T \nabla f(z_{t-1})$ denotes an *error correction term* that removes the previous target. This error correction term is theoretically important for stability [13], as the accumulation of hints can otherwise dominate the true signal. That the original BMG does not feature this error term may explain the instabilities the authors observed when setting too aggressive targets [8]. Since Algorithm 3 does not average its iterates—while Algorithm 4 does—we see that these updates (ignoring $\xi_t$) are identical up to scalar coefficients (that can be controlled for by scaling each $\beta_t$ and each $\tilde{g}_{t+1}$ accordingly). Our next results present formulas for constructing targets in BMG or hints in AO-FTRL so that the two commute.

**Theorem 3.** *Targets in Algorithm 3 and hints in algorithm 4 commute in the following sense.* **BMG → AO-FTRL.** *Let BMG targets $\{z_t\}_{t=1}^T$ by given. A sequence of hints $\{\tilde{g}\}_{t=1}^T$ can be constructed recursively by*

$$\alpha_{t+1}\tilde{g}_{t+1} = D\varphi(\bar{x}_{t-1}, w_t)^T(\nabla\mu(\bar{x}_t) - \nabla\mu(z_t) - \alpha_t\nabla f(\bar{x}_t)) + \alpha_t\tilde{g}_t, \qquad t \in [T], \qquad (11)$$

*so that interior updates for Algorithm 4 are given by*

$$w_{t+1} = \frac{\beta_t}{\beta_{t-1}}w_t - \beta_t\left(\nabla\mu(z_t) - \nabla\mu(\bar{x}_t)\right).$$

**AO-FTRL → BMG.** *Conversely, assume a sequence $\{\tilde{y}_t\}_{t=1}^T$ are given, each $\tilde{y}_t \in \mathbb{R}^n$. If $\mu$ strictly convex, a sequence of BMG targets $\{z_t\}_{t=1}^T$ can be constructed recursively by*

$$z_t = \nabla\mu^{-1}\left(\nabla\mu(x_t) - (\alpha_{t+1}\tilde{y}_{t+1} + \alpha_t\nabla f(x_t))\right) \qquad t \in [T],$$

*so that BMG updates in Eq. 10 are given by*

$$w_{t+1} = w_t - \beta_t\left(\alpha_{t+1}\tilde{g}_{t+1} + \alpha_t(D\varphi(\bar{x}_{t-1}, w_t)^T\nabla f(\bar{x}_t) - \tilde{g}_t)\right),$$

*where each $\tilde{g}_{t+1}$ is given by*

$$\alpha_{t+1}\tilde{g}_{t+1} = \alpha_{t+1}D\varphi(x_{t-1}, w_t)^T\tilde{y}_{t+1} + \alpha_t\tilde{g}_t.$$

Proof; see Appendix E. As an immediate consequence of this, we can apply our optimistic meta-gradient analysis (Theorem 2) to BMG to obtain a rate of convergence. This is captured in the following corollary.

**Corollary 2.** *Let each $\tilde{g}_{t+1} = D\varphi(\bar{x}_{t-1}, w_t)^T\tilde{y}_{t+1}$, for some $\tilde{y}_{t+1} \in \mathbb{R}^n$. If each $\tilde{y}_{t+1}$ is a better predictor of the next gradient than $\nabla f(\bar{x}_{t-1})$, in the sense that*

$$\|D\varphi(\bar{x}_{t-2}, w_{t-1})^T\tilde{y}_t - D\varphi(\bar{x}_{t-1}, w_t)^T\nabla f(\bar{x}_t)\|_* \le \tilde{\lambda}\|\nabla f(\bar{x}_t) - \nabla f(\bar{x}_{t-1})\|_*,$$

*then Algorithm 4 guarantees convergence at a rate $O(\tilde{\lambda}/T^2)$.*

In other words, for certain choices of targets, BMG yields accelerated rates of convergence.

# 7 Experiments

In this section, we detail experiments borne out to test our theoretical predictions. Section 7.1 tests predictions about meta-gradients without optimism, by comparing meta-learned variants of standard optimisers to their non-meta-learned counterparts. Section 7.2 test predictions about optmistic meta-gradients, by comparing an optimistic meta-gradient approach to gradient descent.

## 7.1 Convex Quadratic Experiments

**Loss function.** We consider the problem of minimising a convex quadratic loss functions $f : \mathbb{R}^2 \to \mathbb{R}$ of the form $f(x) = x^TQx$ for some $Q$ that is sampled such that it is ill-conditioned (see Appendix B for details).

**Protocol.** Given that the solution is always $(0, 0)$, this experiment revolves around understanding how different algorithms deal with curvature. Given symmetry in the solution and ill-conditioning, we fix the initialisation to $x_0 = (4, 4)$ and run each algorithm for 100 iterations. For each $Q$ and each algorithm, we sweep over the learning rate, decay rate, and the initialization of $w$ (see Table 2 for values) and report results for the best performing hyper parameters.

**Results.** We report the learning curves for the best hyper-parameter choice for 5 randomly sampled problems in the top row of Figure 2 (columns correspond to different Q). We also study the sensitivity of each algorithm to the learning rate in the bottom row Figure 2. For each learning rate, we report the cumulative loss during training. While baselines are relatively insensitive to hyper-parameter choice, meta-learned improve for certain choices, but are never worse than baselines.

## 7.2 Imagenet Experiments

**Protocol.** We train a 50-layer ResNet following a standard protocol (Appendix C) with SGD as the baseline optimiser. We compare SGD to a variant that meta-learns an element-wise learning rate online, i.e. $(x, w) \mapsto w \odot \nabla f(x)$. We sweep over the learning rate (for SGD) or meta-learning rate and report results for the best hyper-parameter over three independent runs.

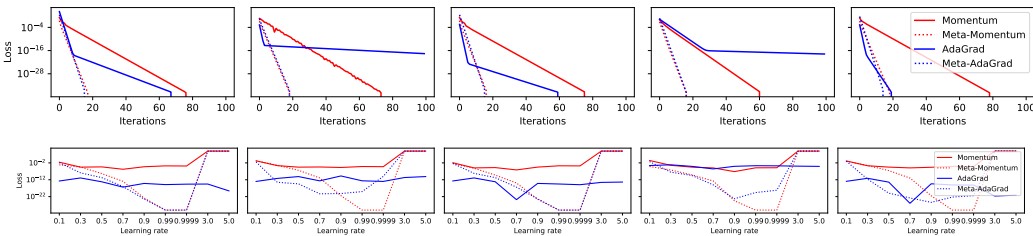

Figure 2: Convex Quadratic. We generate convex quadratic loss functions with ill-conditioning and compare gradient descent with momentum and AdaGrad to meta-learning variants. Meta-Momentum uses $\varphi : (x, w) \mapsto w \odot \nabla f(x)$ while Meta-AdaGrad uses $\varphi : (x, w) \mapsto \nabla f(x)/\sqrt{w}$, where division is element-wise. *Top:* loss per iteration for randomly sampled loss functions. *Bottom:* cumulative loss (regret) at the end of learning as a function of learning rate; details in Appendix B.

**Standard meta-learning.** In the standard meta-learning setting, we apply the update rule once before differentiating w.r.t. the meta-parameters. That is, the meta-update takes the form $w_{t+1} = w_t - \beta \nabla h_t(w_t)$, where $h_t = f(x_t + w_t \odot \nabla f(x_t))$. Because the update rule is linear in $w$, we can compute the meta-gradient analytically: $\nabla h_t(w_t) = \nabla_w f(x + \varphi(x, w)) = D\varphi(x, w)^T \nabla f(x') = \nabla f(x) \odot \nabla f(x')$, where $x' = x + \varphi(x, w)$. Hence, we can compute the meta-updates in Algorithm 1 manually as $w_{t+1} = \max\{w_t - \beta \nabla f(x_t) \odot \nabla f(x_{t+1}), 0.\}$, where we introduce the $\max$ operator on an element-wise basis to avoid negative learning rates. Empirically, this was important to stabilize training.

**Optimistic meta-learning.** For optimistic meta-learning, we proceed much in the same way, but include a gradient prediction $\tilde{g}_{t+1}$. For our prediction, we use the previous gradient, $\nabla f(x_{t+1})$, as our prediction. This yields meta-updates of the form $w_{t+1} = \max\{w_t - \beta \nabla f(x_{t+1}) \odot (\nabla f(x_{t+1}) + \nabla f(x_t)) - \nabla f(x_t) \odot \nabla f(x_t), 0.\}$,.

**Results.** We report Top-1 accuracy on the held-out test set as a function of training steps in Figure 1. Tuning the learning rate does not yield any statistically significant improvements under standard meta-learning. However, with optimistic meta-learning, we obtain a significant acceleration as well as improved final performance, increasing the mean final top-1 accuracy from $72\%$ to $75\%$.

## 8 Conclusion

This paper explores a connection between convex optimisation and meta-learning. We construct an algorithm for convex optimisation that aligns as closely as possible with how meta-learning is done in practice. Meta-learning introduces a transformation and we study the effect this transformation has on the rate of convergence. We find that, while a meta-learned update rule cannot generate a better dependence on the horizon $T$, it can improve upon classical optimisation up to a constant factor.

An implication of our analysis is that for meta-learning to achieve acceleration, it is important to introduce some form of optimism. From a classical optimisation point of view, such optimism arises naturally by providing the meta-learner with hints. If hints are predictive of the learning dynamics these can lead to significant acceleration. We show that the recently proposed BMG method provides a natural avenue to incorporate optimism in practical application of meta-learning. Because targets in BMG and hints in optimistic online learning commute, our results provide first rigorous proof of convergence for BMG, while providing a general condition under which optimism in BMG yields accelerated learning.

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
