# Appendix

## A Notation

Table 1: Notation

| | |
|---|---|
| **Indices** | |
| $t$ | Iteration index: $t \in \{1, ..., T\}$. |
| $T$ | Total number of iterations. |
| $[T]$ | The set $\{1, 2, \ldots, T\}$. |
| $i$ | Component index: $x^i$ is the $i$th component of $x = (x^1, \ldots, x^n)$. |
| $\alpha_{a:b}$ | Sum of weights: $\alpha_{a:b} = \sum_{s=a}^{b} \alpha_s$ |
| $x_{a:b}$ | Weighted sum: $x_{a:b} = \sum_{s=a}^{b} \alpha_s x_s$ |
| $\bar{x}_{a:b}$ | Weighted average: $\bar{x}_{a:b} = x_{a:b}/\alpha_{a:b}$ |
| | |
| **Parameters** | |
| $x^* \in \mathcal{X}$ | Minimiser of $f$. |
| $x_t \in \mathcal{X}$ | Parameter at time $t$ |
| $\bar{x}_t \in \mathcal{X}$ | Moving average of $\{x_s\}_{s=1}^t$ under weights $\{\alpha_s\}_{s=1}^t$. |
| $\rho_t \in (0, \infty)$ | Moving average coefficient $\alpha_t/\alpha_{1:t}$. |
| $w_t \in \mathcal{W}$ | Meta parameters |
| $w^* \in \mathcal{X}$ | $w \in \mathcal{W}$ that retains regret with smallest norm $\|w\|$. |
| $\alpha_t \in (0, \infty)$ | Weight coefficients |
| $\beta_t \in (0, \infty)$ | Meta-learning rate |
| | |
| **Maps** | |
| $f : \mathcal{X} \to \mathbb{R}$ | Objective function |
| $\|\cdot\| : \mathcal{X} \to \mathbb{R}$ | Norm on $\mathcal{X}$. |
| $\|\cdot\|_* : \mathcal{X}^* \to \mathbb{R}$ | Dual norm of $\|\cdot\|$. |
| $h_t : \mathcal{W} \to \mathbb{R}$ | Online loss faced by the meta learner |
| $R^x(T)$ | Regret of $\{x_t\}_{t=1}^T$ against $x^*$: $R^x(T) \coloneqq \sum_{t=1}^T \alpha_t \langle \nabla f(\bar{x}_t), x_t - x^* \rangle$. |
| $R^w(T)$ | $R^w(T) \coloneqq \sum_{t=1}^T \alpha_t \langle \nabla f(\bar{x}_t), \varphi(\bar{x}_{t-1}, w_t) - \varphi(\bar{x}_{t-1}, w^*) \rangle$. |
| $\varphi : \mathbb{R}^n \times \mathbb{R}^m \to \mathbb{R}^n$ | Generic update rule used in practice |
| $D\varphi(x, \cdot) : \mathbb{R}^m \to \mathbb{R}^{n \times m}$ | Jacobian of $\varphi$ w.r.t. its second argument, evaluated at $x \in \mathbb{R}^n$. |
| $\varphi : \mathcal{X} \times \mathcal{W} \to \mathcal{X}$ | Update rule in convex setting |
| $D\varphi(x, \cdot) : \mathcal{W} \to \mathbb{R}^{n \times m}$ | Jacobian of $\varphi$ w.r.t. its second argument, evaluated at $x \in \mathcal{X}$. |
| $B^\mu : \mathbb{R}^n \times \mathbb{R}^n \to [0, \infty)$ | Bregman divergence under $\mu : \mathbb{R}^n \to \mathbb{R}$. |
| $\mu : \mathbb{R}^n \to \mathbb{R}$ | Convex distance generating function. |

Table 2: Hyper-parameter sweep on Convex Quadratics. All algorithms are tuned for learning rate and initialisation of $w$. Baselines are tuned for decay rate; meta-learned variant are tuned for the meta-learning rate.

| | |
|---|---|
| Learning rate | [.1, .3, .7, .9, 3., 5.] |
| $w$ init scale | [0., 0.3, 1., 3., 10., 30.] |
| Decay rate / Meta-learning rate | [0.001, 0.003, 0.01, .03, .1, .3, 1., 3., 10., 30.] |

Table 3: Hyper-parameter sweep on Imagenet.

| | |
|---|---|
| (Meta-)learning rate | [0.001, 0.01, 0.02, 0.05, 0.1] |

## B  Convex Quadratic Experiments

**Loss function.**  We consider the problem of minimising a convex quadratic loss functions $f :$ $\mathbb{R}^2 \rightarrow \mathbb{R}$ of the form $f(x) = x^T Q x$, where $Q$ is randomly sampled as follows. We sample a random orthogonal matrix $U$ from the Haar distribution `scipy.stats.ortho_group`. We construct a diagonal matrix of eigenvalues, ranked smallest to largest, with $\lambda_i = i^2$. Hence, the first dimension has an eigenvalue 1 and the second dimension has eigenvalue 4. The matrix $Q$ is given by $U^T \operatorname{diag}(\lambda_1, \ldots, \lambda_n) U$.

## C  Imagenet Experiments

**Protocol.**  We train a 50-layer ResNet following the Haiku example, available at `https://github.com/deepmind/dm-haiku/blob/main/examples/imagenet`. We modify the default setting to run with SGD.

## D  Proofs

This section provides complete proofs. We restate the results for convenience.

**Lemma 1.** *Given $f$, $\{\alpha_t\}_{t=1}^T$, and $\{x_t\}_{t=1}^T$, if $\varphi$ preserves regret, then*

$$R^x(T) = \sum_{t=1}^T \alpha_t \langle \nabla f(\bar{x}_t), x_t - x^* \rangle \leq \sum_{t=1}^T \alpha_t \langle \nabla f(\bar{x}_t), \varphi(\bar{x}_{t-1}, w_t) - \varphi(\bar{x}_{t-1}, w^*) \rangle = R^w(T).$$

*Proof.* Starting from $R^x$ in Eq. 7, if the update rule preserves regret, there exists $w^* \in \mathcal{W}$ for which

$$
\begin{aligned}
R^x(T) &= \sum_{t=1}^T \alpha_t \langle \nabla f(\bar{x}_T), \varphi(\bar{x}_{t-1}, w_t) - x^* \rangle \\
&= \sum_{t=1}^T \alpha_t \langle \nabla f(\bar{x}_T), \varphi(\bar{x}_{t-1}, w_t) - \varphi(\bar{x}_{t-1}, w^*) \rangle + \sum_{t=1}^T \alpha_t \langle \nabla f(\bar{x}_T), \varphi(\bar{x}_{t-1}, w^*) - x^* \rangle \\
&\leq \sum_{t=1}^T \alpha_t \langle \nabla f(\bar{x}_T), \varphi(\bar{x}_{t-1}, w_t) - \varphi(\bar{x}_{t-1}, w^*) \rangle = R^w(T),
\end{aligned}
$$

since $w^*$ is such that $\sum_{t=1}^T \alpha_t \langle \nabla f(\bar{x}_T), \varphi(\bar{x}_{t-1}, w^*) - x^* \rangle \leq 0$. ∎

**Theorem 1.** *Let $\varphi$ preserve regret and assume Algorithm 2 satisfy the assumptions in Section 2. Then*

$$
\begin{aligned}
f(\bar{x}_T) - f(x^*) \leq & \frac{1}{\alpha_{1:T}} \left( \frac{\|w^*\|^2}{\beta} + \sum_{t=1}^T \frac{\lambda \beta \alpha_t^2}{2} \|\nabla f(\bar{x}_t)\|_*^2 \right. \\
& \left. - \frac{\alpha_t}{2L} \|\nabla f(\bar{x}_t) - \nabla f(x^*)\|_*^2 - \frac{\alpha_{1:t-1}}{2L} \|\nabla f(\bar{x}_{t-1}) - \nabla f(\bar{x}_t)\|_*^2 \right).
\end{aligned}
$$

*If $x^*$ is a global minimiser of $f$, setting $\alpha_t = 1$ and $\beta = \frac{1}{\lambda L}$ yields $f(\bar{x}_T) - f(x^*) \leq \frac{\lambda L \operatorname{diam}(\mathcal{W})}{T}$.*

*Proof.* Since $\varphi$ preserves regret, by Lemma 1, the regret term $R^x(T)$ in Eq. 6 is upper bounded by $R^w(T)$. We therefore have

$$f(\bar{x}_T) - f(x^*) \leq$$
$$\frac{1}{\alpha_{1:T}} \left( R^w(T) - \frac{\alpha_t}{2L} \|\nabla f(\bar{x}_t) - \nabla f(x^*)\|_*^2 - \frac{\alpha_{1:t-1}}{2L} \|\nabla f(\bar{x}_{t-1}) - \nabla f(\bar{x}_t)\|_*^2 \right). \tag{12}$$

Next, we need to upper-bound $R^w(T)$. Since, $R^w(T) = \sum_{t=1}^T \alpha_t \langle \nabla f(\bar{x}_T), \varphi(\bar{x}_{t-1}, w_t) - \varphi(\bar{x}_{t-1}, w^*) \rangle$, the regret of $\{w_t\}_{t=1}^T$ is defined under loss functions $h_t : \mathcal{W} \to \mathbb{R}$ given by $h_t = \alpha_t \langle \nabla f(\bar{x}_T), \varphi(\bar{x}_{t-1}, w)) \rangle$. By assumption of convexity in $\varphi$, each $h_t$ is convex in $w$. Hence, the regret under $\{\alpha_t h_t\}_{t=1}^T$ can be upper bounded by the regret under the linear losses $\{\alpha_t \langle \nabla h_t(w_t), \cdot \rangle\}_{t=1}^T$. These linear losses correspond to the losses used in the meta-update in Eq. 3. Since the meta-update is an instance of FTRL, we may upper-bound $R^w(T)$ by Eq. 5 with each $\tilde{g}_t = 0$. Putting this together along with smoothness of $\varphi$,

$$R^x(T) \leq R^w(T)$$
$$= \sum_{t=1}^T \alpha_t \langle \nabla f(\bar{x}_T), \varphi(\bar{x}_{t-1}, w_t) - \varphi(\bar{x}_{t-1}, w^*) \rangle$$
$$\leq \sum_{t=1}^T \alpha_t \langle \nabla h_t(w_t), w_t - w^* \rangle$$
$$\leq \frac{\|w^*\|^2}{\beta} + \frac{\beta}{2} \sum_{t=1}^T \alpha_t^2 \|\nabla h_t(w_t)\|_*^2$$
$$= \frac{\|w^*\|^2}{\beta} + \frac{\beta}{2} \sum_{t=1}^T \alpha_t^2 \|D\varphi(\bar{x}_{t-1}, w_t)^T \nabla f(\bar{x}_t)\|_*^2$$
$$\leq \frac{\|w^*\|^2}{\beta} + \frac{\lambda\beta}{2} \sum_{t=1}^T \alpha_t^2 \|\nabla f(\bar{x}_t)\|_*^2. \tag{13}$$

Putting Eq. 12 and Eq. 13 together gives the stated bound. Next, if $x^*$ is the global optimiser, $\nabla f(x^*) = 0$ by first-order condition. Setting $\beta = 1/(L\lambda)$ and $\alpha_t = 1$ means the first two norm terms in the summation cancel. The final norm term in the summation is negative and can be ignored. We are left with $f(\bar{x}_T) - f(x^*) \leq \frac{\lambda L \|w^*\|^2}{T} \leq \frac{\lambda L \operatorname{diam}(\mathcal{W})}{T}$. ∎

**Theorem 2** *Let $\varphi$ preserve regret and assume Algorithm 4 satisfy the assumptions in Section 2. Then*

$$f(\bar{x}_T) - f(x^*) \leq \frac{1}{\alpha_{1:T}} \left( \frac{\|w^*\|^2}{\beta_T} + \sum_{t=1}^T \frac{\alpha_t^2 \beta_t}{2} \|D\varphi(\bar{x}_{t-1}, w_t)^T \nabla f(\bar{x}_t) - \tilde{g}_t\|_*^2 \right.$$
$$\left. - \frac{\alpha_t}{2L} \|\nabla f(\bar{x}_t) - \nabla f(x^*)\|_*^2 - \frac{\alpha_{1:t-1}}{2L} \|\nabla f(\bar{x}_{t-1}) - \nabla f(\bar{x}_t)\|_*^2 \right).$$

*Proof.* The proof follows the same lines as that of Theorem 1. The only difference is that the regret of the $\{w_t\}_{t=1}^T$ sequence can be upper bounded by $\frac{\|w^*\|^2}{\beta_T} + \frac{1}{2} \sum_{t=1}^T \alpha_t^2 \beta_t \|\nabla h_t(w_t) - \tilde{g}_t\|_*^2$ instead of $\frac{\|w^*\|^2}{\beta_T} + \frac{1}{2} \sum_{t=1}^T \alpha_t^2 \beta_t \|\nabla h_t(w_t)\|_*^2$, as per the AO-FTRL regret bound in Eq. 5. ∎

**Corollary 1.** *Let each $\tilde{g}_{t+1} = D\varphi(\bar{x}_{t-1}, w_t)^T \nabla f(\bar{x}_t)$. Assume that $\varphi$ satisfies*

$$\left\| D\varphi(x', w)^T \nabla f(x) - D\varphi(x'', w')^T \nabla f(x') \right\|_*^2 \leq \tilde{\lambda} \left\| \nabla f(x') - \nabla f(x) \right\|_*^2$$

*for all $x'', x', x \in \mathcal{X}$ and $w, w' \in \mathcal{W}$, for some $\tilde{\lambda} > 0$. If each $\alpha_t = t$ and $\beta_t = \frac{t-1}{2t\tilde{\lambda}L}$, then $f(\bar{x}_T) - f(x^*) \leq \frac{4\tilde{\lambda}L \operatorname{diam}(\mathcal{W})}{T^2 - 1}$.*

*Proof.* Plugging in the choice of $\tilde{g}_t$ and using that

$$\left\|D\varphi(\bar{x}_{t-1}, w_t)^T \nabla f(\bar{x}_t) - D\varphi(x_{t-2}, w_{t-1})^T \nabla f(\bar{x}_{t-1})\right\|_*^2 \leq \tilde{\lambda} \left\|\nabla f(\bar{x}_{t-1}) - \nabla f(\bar{x}_t)\right\|_*^2,$$

the bound in Theorem 2 becomes

$$f(\bar{x}_T) - f(x^*) \leq \frac{1}{\alpha_{1:T}} \left( \frac{\|w^*\|^2}{\beta_T} + \frac{1}{2} \sum_{t=1}^{T} \left( \tilde{\lambda} \alpha_t^2 \beta_t - \frac{\alpha_{1:t-1}}{L} \right) \|\nabla f(\bar{x}_t) - \nabla f(\bar{x}_{t-1})\|_*^2 \right),$$

where we drop the negative terms $\|\nabla f(\bar{x}_t) - \nabla f(x^*)\|_*^2$. Setting $\alpha_t = t$ yields $\alpha_{1:t-1} = \frac{(t-1)t}{2}$, while setting $\beta_t = \frac{t-1}{2t\tilde{\lambda}L}$ means $\tilde{\lambda}\alpha_t^2 \beta_t = \frac{(t-1)t}{2L}$. Hence, $\tilde{\lambda}\alpha_t^2 \beta_t - \alpha_{1:t-1}/L$ cancels and we get

$$f(\bar{x}_T) - f(x^*) \leq \frac{\|w^*\|^2}{\beta_T \alpha_{1:T}} = \frac{4\|w^*\|^2 \tilde{\lambda}L}{(T-1)(T+1)} \leq \frac{4\tilde{\lambda}L \operatorname{diam}(\mathcal{W})}{(T-1)(T+1)} = \frac{4\tilde{\lambda}L \operatorname{diam}(\mathcal{W})}{T^2 - 1}.$$

∎

# E   BMG as an instance of Optimism

To prove Theorem 3, we begin by showing that AO-FTRL reduces to a particular form.

**Lemma 2.** *Consider Algorithm 4. Given online losses $h_t : \mathcal{W} \to \mathbb{R}$ defined by $\{\langle D\varphi(\bar{x}_{t-1}, w_t)^T \nabla f(\bar{x}_t), \cdot \rangle\}_{t=1}^T$ and hint functions $\{\langle \tilde{g}_t, \cdot, \rangle\}_{t=1}^T$, with each $\tilde{g}_t \in \mathbb{R}^m$. If $\|\cdot\| = (1/2)\|\cdot\|_2$, an interior solution to Eq. 9 is given by*

$$w_{t+1} = \frac{\beta_t}{\beta_{t-1}} w_t - \beta_t \left( \alpha_{t+1} \tilde{g}_{t+1} + \alpha_t (D\varphi(\bar{x}_{t-1}, w_t)^T \nabla f(\bar{x}_t) - \tilde{g}_t) \right).$$

*Proof.* By direct computation:

$$w_{t+1} = \underset{w \in \mathcal{W}}{\arg\min} \left( \alpha_{t+1} \langle \tilde{g}_{t+1}, w \rangle + \sum_{s=1}^{t} \alpha_s \langle D\varphi(\bar{x}_{s-1}, w_s)^T \nabla f(\bar{x}_s), w \rangle + \frac{1}{2\beta_t} \|w\|_2^2 \right)$$

$$= -\beta_t \left( \alpha_{t+1} \tilde{g}_{t+1} + \sum_{s=1}^{t} \alpha_t D\varphi(\bar{x}_{s-1}, w_s)^T \nabla f(\bar{x}_s) \right)$$

$$= -\beta_t \left( \alpha_{t+1} \tilde{g}_{t+1} + \alpha_t D\varphi(\bar{x}_{t-1}, w_t)^T \nabla f(\bar{x}_t) + \left( \sum_{s=1}^{t-1} \alpha_t D\varphi(\bar{x}_{s-1}, w_s)^T \nabla f(\bar{x}_s) \right) \right)$$

$$= -\beta_t \left( \alpha_{t+1} \tilde{g}_{t+1} + \alpha_t (D\varphi(\bar{x}_{t-1}, w_t)^T \nabla f(\bar{x}_t) - \tilde{g}_t) \right)$$

$$\quad - \beta_t \left( \alpha_t \tilde{g}_t + \sum_{s=1}^{t-1} \alpha_t D\varphi(\bar{x}_{s-1}, w_s)^T \nabla f(\bar{x}_s) \right)$$

$$= \frac{\beta_t}{\beta_{t-1}} w_t - \beta_t \left( \alpha_{t+1} \tilde{g}_{t+1} + \alpha_t (D\varphi(\bar{x}_{t-1}, w_t)^T \nabla f(\bar{x}_t) - \tilde{g}_t) \right).$$

∎

AO-FTRL includes a decay rate $\beta_t/\beta_{t-1}$; this decay rate can be removed by instead using optimistic online mirror descent [24, 12]—to simplify the exposition we consider only FTRL-based algorithms in this paper. An immediate implication of Lemma 2 is the error-corrected version of BMG.

**Corollary 3.** *Setting $\tilde{g}_{t+1} = D\varphi(\bar{x}_{t-1}, w_t)^T \tilde{y}_{t+1}$ for some $\tilde{y}_{t+1} \in \mathbb{R}^n$ yields an error-corrected version of the BMG meta-update in Eq. 10. Specifically, the meta-updates in Lemma 2 becomes*

$$w_{t+1} = \frac{\beta_t}{\beta_{t-1}} w_t - \underbrace{\beta_t D\varphi(\bar{x}_{t-1}, w_t)^T (\alpha_{t+1} \tilde{y}_{t+1} + \alpha_t \nabla f(\bar{x}_t))}_{\text{BML update}} + \underbrace{\beta_t \alpha_t D\varphi(\bar{x}_{t-2}, w_{t-1})^T \tilde{y}_t}_{\text{FTRL error correction}}.$$

*Proof.* Follows immediately by substituting for each $\tilde{g}_{t+1}$ in Lemma 2. ∎

**Theorem 3** *Targets in Algorithm 3 and hints in algorithm 4 commute in the following sense.* **BMG → AO-FTRL.** *Let BMG targets $\{z_t\}_{t=1}^T$ by given. A sequence of hints $\{\tilde{g}\}_{t=1}^T$ can be constructed recursively by*

$$\alpha_{t+1}\tilde{g}_{t+1} = D\varphi(\bar{x}_{t-1}, w_t)^T(\nabla\mu(\bar{x}_t) - \nabla\mu(z_t) - \alpha_t\nabla f(\bar{x}_t)) + \alpha_t\tilde{g}_t, \qquad t \in [T], \qquad (14)$$

*so that interior updates for Algorithm 4 are given by*

$$w_{t+1} = \frac{\beta_t}{\beta_{t-1}}w_t - \beta_t\left(\nabla\mu(z_t) - \nabla\mu(\bar{x}_t)\right).$$

**AO-FTRL → BMG.** *Conversely, assume a sequence $\{\tilde{y}_t\}_{t=1}^T$ are given, each $\tilde{y}_t \in \mathbb{R}^n$. If $\mu$ strictly convex, a sequence of BMG targets $\{z_t\}_{t=1}^T$ can be constructed recursively by*

$$z_t = \nabla\mu^{-1}\left(\nabla\mu(x_t) - (\alpha_{t+1}\tilde{y}_{t+1} + \alpha_t\nabla f(x_t))\right) \qquad t \in [T],$$

*so that BMG updates in Eq. 10 are given by*

$$w_{t+1} = w_t - \beta_t\left(\alpha_{t+1}\tilde{g}_{t+1} + \alpha_t(D\varphi(\bar{x}_{t-1}, w_t)^T\nabla f(\bar{x}_t) - \tilde{g}_t)\right),$$

*where each $\tilde{g}_{t+1}$ is given by*

$$\alpha_{t+1}\tilde{g}_{t+1} = \alpha_{t+1}D\varphi(x_{t-1}, w_t)^T\tilde{y}_{t+1} + \alpha_t\tilde{g}_t.$$

*Proof.* First, consider BMG → AO-FTRL. First note that $\tilde{g}_1$ is never used and can thus be chosen arbitrarily—here, we set $\tilde{g}_1 = 0$. For $w_2$, Lemma 2 therefore gives the interior update

$$w_2 = \frac{\beta_2}{\beta_1}w_1 - \beta_1(\alpha_2\tilde{g}_2 + \alpha_1 D\varphi(\bar{x}_0, w_1)^T\nabla f(\bar{x}_1)).$$

Since the formula for $\tilde{g}_2$ in Eq. 11 only depends on quantities with iteration index $t = 0, 1$, we may set $\alpha_2\tilde{g}_t = D\varphi(\bar{x}_0, w_1)^T(\nabla\mu(\bar{x}_1) - \nabla\mu(z_t) - \alpha_t\nabla f(\bar{x}_1))$. This gives the update

$$w_2 = \frac{\beta_2}{\beta_1}w_1 - \beta_1 D\varphi(\bar{x}_0, w_1)^T(\nabla\mu(\bar{x}_1) - \nabla\mu(z_1)).$$

Now assume the recursion holds up to time $t$. As before, we may choose $\alpha_{t+1}\tilde{g}_{t+1}$ according to the formula in Eq. 11 since all quantities on the right-hand side depend on quantities computed at iteration $t$ or $t - 1$. Substituting this into Lemma 2, we have

$$
\begin{aligned}
w_{t+1} &= \frac{\beta_t}{\beta_{t-1}}w_t - \beta_t\left(\alpha_{t+1}\tilde{g}_{t+1} + \alpha_t(D\varphi(\bar{x}_{t-1}, w_t)^T\nabla f(\bar{x}_t) - \tilde{g}_t)\right) \\
&= \frac{\beta_t}{\beta_{t-1}}w_t - \beta_t\left(D\varphi(\bar{x}_{t-1}, w_t)^T(\nabla\mu(\bar{x}_t) - \nabla\mu(z_t) - \alpha_t\nabla f(\bar{x}_t)) + \alpha_t\tilde{g}_t \right. \\
&\quad \left. + \alpha_t(D\varphi(\bar{x}_{t-1}, w_t)^T\nabla f(\bar{x}_t) - \tilde{g}_t)\right) \\
&= \frac{\beta_t}{\beta_{t-1}}w_t - \beta_t D\varphi(\bar{x}_{t-1}, w_t)^T(\nabla\mu(\bar{x}_t) - \nabla\mu(z_t)).
\end{aligned}
$$

AO-FTRL → BMG. The proof in the other direction follows similarly. First, note that for $\mu$ strictly convex, $\nabla\mu$ is invertible. Then, $z_1 = \nabla\mu^{-1}(\nabla\mu(x_1) - (\alpha_2\tilde{y}_2 + \alpha_1\nabla f(x_1)))$. This target is permissible since $x_1$ is already computed and $\{\tilde{y}_t\}_{t=1}^T$ is given. Substituting this into the BMG meta-update in Eq. 10, we find

$$
\begin{aligned}
w_2 &= w_1 - \beta_1 D\varphi(x_0, w_1)^T(\nabla\mu(x_1) - \nabla\mu(\nabla\mu^{-1}(\nabla\mu(x_1) - (\alpha_2\tilde{y}_2 + \alpha_1\nabla f(x_1))))) \\
&= w_1 - \beta_1 D\varphi(x_0, w_1)^T(\alpha_2\tilde{y}_2 + \alpha_1\nabla f(x_1)) \\
&= w_1 - \beta_1\left(\alpha_2\tilde{g}_2 + \alpha_1(D\varphi(\bar{x}_0, w_1)^T\nabla f(\bar{x}_1) - \tilde{g}_1)\right),
\end{aligned}
$$

where the last line uses that $\tilde{g}_2$ is defined by $\alpha_2\tilde{g}_2 - \alpha_1\tilde{g}_1 = D\varphi(\bar{x}_0, w_1)^T\tilde{y}_2$ and $\tilde{g}_1$ is arbitrary. Again, assume the recursion holds to time $t$. We then have

$$
\begin{aligned}
w_{t+1} &= w_t - \beta_t D\varphi(x_{t-1}, w_t)^T\left(\nabla\mu(x_t) - \nabla\mu(z_t)\right) \\
&= w_t - \beta_t D\varphi(x_{t-1}, w_t)^T(\nabla\mu(x_t) \\
&\quad - \nabla\mu(\nabla\mu^{-1}(\nabla\mu(x_t) - (\alpha_{t+1}\tilde{y}_{t+1} + \alpha_t\nabla f(x_t))))) \\
&= w_t - \beta_t D\varphi(x_{t-1}, w_t)^T(\alpha_{t+1}\tilde{y}_{t+1} + \alpha_t\nabla f(x_t)) \\
&= w_t - \beta_t(\alpha_{t+1}\tilde{g}_{t+1} + \alpha_t(D\varphi(x_{t-1}, w_t)^T\nabla f(x_t) - \tilde{g}_t)).
\end{aligned}
$$

$\blacksquare$

**Corollary 2.** *Let each $\tilde{g}_{t+1} = D\varphi(\bar{x}_{t-1}, w_t)^T \tilde{y}_{t+1}$, for some $\tilde{y}_{t+1} \in \mathbb{R}^n$. If each $\tilde{y}_{t+1}$ is a better predictor of the next gradient than $\nabla f(\bar{x}_{t-1})$, in the sense that*

$$\|D\varphi(\bar{x}_{t-2}, w_{t-1})^T \tilde{y}_t - D\varphi(\bar{x}_{t-1}, w_t)^T \nabla f(\bar{x}_t)\|_* \leq \tilde{\lambda}\|\nabla f(\bar{x}_t) - \nabla f(\bar{x}_{t-1})\|_*,$$

*then Algorithm 4 guarantees convergence at a rate $O(\tilde{\lambda}/T^2)$.*

*Proof.* The proof follows the same argument as Corollary 1. ∎