# OpenReview forum: "Optimistic Meta-Gradients"
_NeurIPS.cc/2023/Conference — NeurIPS 2023 poster_

### Official Review · Reviewer_fhwU · 2023-06-23

**Soundness:** 3 good
**Presentation:** 3 good
**Contribution:** 2 fair
**Rating:** 6
**Confidence:** 3

**Summary:**

This paper studies a connection between optimization and meta-learning. For the case of a single task, it shows an equivalence between GD with momentum and GBML, and another equivalence between GD with Nesterov acceleration and the recent Bootstrapped Meta-Gradient algorithm. Theoretical analyses are done for these algorithm, showing that GBML speeds up convergence by a constant factor but is not able to improve on the $O(1/T)$ rate, while BMG is able to improve the rate to $O(1/T^2)$. Experiments are done on quadratic minimization and ImageNet image classification that confirm the theory.

**Strengths:**

1. The mathematical argument in the paper is very clearly presented, e.g. the authors roughly outline proof techniques.
2. Overall the paper appears to be sound, although I did not check the math. Experiments were done on both "toy" examples and a more realistic ImageNet problem.
3. As far as I know, the connection between BMG and GD with Nesterov acceleration is novel.

**Weaknesses:**

In my opinion, the major weakness of the paper is the fact that it only considers the single task setting. However, one of the major motivations behind meta-learning is that information may be shared between tasks in order to improve convergence speeds for all tasks, and the multi-task setting is arguably more popular in the literature. It would greatly strengthen the work to consider the theory for a multi-task setting, although I understand that it might be more appropriate for future work.

Minor:
1. The objective function is convex is a fairly strong assumption.

**Questions:**

Is it fair to search over the range of hyperparameters in the experiments in Section 7.1? Usually when this is done the final results are provided for a held-out test set (a new quadratic function in this case), but this does not seem to be the case here.

**Limitations:**

The authors do not have a broader impact section, and I agree since this work discusses optimization methods. However, they do not discuss the limitations/weaknesses of their work, which I think should be added to the final draft.

---

> ### Author Rebuttal · Authors · 2023-08-07
>
> Thank you for your review, we are glad you found the paper well presented with novel insights. We appreciate that multi-task meta-learning is a common problem setting and one that we do not consider in this paper. While multi-task meta-learning has been fairly extensively analyzed, we are not aware of any theoretical work in the single-task setting (except some papers in the online optimization community on meta-learned learning rates). The goal of this paper is to address this gap, hence the focus on single task online meta-learning.
>
> Notably, as the reviewer points out, rates of acceleration in the multi-task setting rely on transfer across tasks. Reading these papers, one may be tempted to conclude that meta-learning can only achieve acceleration by transferring knowledge across tasks. We believe this to be a commonly held belief in the community.  Significantly, our paper proves the *opposite* to be true: meta-learning can achieve acceleration *without* the need for multiple tasks! We believe this to be a fundamental insight that contravenes current mainstream thinking within the meta-learning community.
>
> Fundamentally, we show that meta-learning is the same as standard optimization, just non-linearly transformed. For us, this was an eye-opening discovery that has changed how we think about meta-learning. We hope that this paper can help others in the field to similarly deepen their understanding of meta-learning.
>
> On the empirical results in Section 7.1: this is a fair question. Proofs of accelerated convergence always depend on an optimal choice of learning rates. Hence, as our purpose is to verify theoretical predictions, the relevant empirical study is to compare algorithms under optimal learning rates. This can be seen as a comparison of ‘best case performance’; as long as all algorithms are tuned similarly, this is still a fair comparison.
>
> On limitations; the main limitation of our work is the assumption of convexity. We discuss this briefly in Section 2, but will expand this discussion in an updated version of the manuscript. If the reviewer would like to see other limitations discussed, please do let us know.

---

> > ### Comment · Reviewer_fhwU · 2023-08-15
> > **Thanks to the authors for the rebuttal**
> >
> > After reading the other reviews and rebuttals, I feel that the authors have adequately addressed my concerns and I have decided to raise my score from 5 to 6.

---

### Official Review · Reviewer_nSnp · 2023-07-01

**Soundness:** 4 excellent
**Presentation:** 3 good
**Contribution:** 3 good
**Rating:** 7
**Confidence:** 3

**Summary:**

This paper shed an interesting perspective on meta-learning by studying the connection between gradient-based meta-learning and convex optimisation in the single task setting. It shows that meta-gradients contain gradient descent with momentum and Nesterov Acceleration as special cases. Furthermore, gradient-based meta-learning can be understood as a non-linear transformation of an underlying optimisation method. The authors establish the rates of convergence for meta-gradients in convex settings. For meta-learning to achieve acceleration of convergence, some form of optimization is needed. Specifically, it provides the first rigorous proof of BMG with acceleration rate of convergence, and highlights the underlying mechanics that enable faster learning with BMG.


**Strengths:**

This paper provides us a novel view on meta-learning by studying the connection between gradient-based meta-learning and convex optimisation. It reveals that gradient descent with momentum and Nesterov Acceleration are special cases of gradient-based meta-learning with different update rules. This paper is of high technical quality. Based on the new understanding for meta-learning as a non-linear transformation of an underlying optimisation method, it establishes theoretical analysis and proof of the convergence rates of meta-gradients in convex settings. The theoretical analysis has insightful implications that we are able to further accelerate the rate of convergence with some form of optimization. As a result, the recently proposed BMG method is proven to achieve an accelerating convergence rate. These results are important and significant, providing us with new deep understanding about gradient-based meta-learning, and theoretical guidance to design fast meta-learning algorithms with update rules. Last, the paper is well organized and clearly written.


**Weaknesses:**

This is a theory paper, it could be more impactful if the authors may add more experiment results for large scale problems either in convex optimization or deep learning settings besides ImageNet.


**Questions:**

Curious to know what is the computational cost of Optimistic meta-learning compared to SGD in the ImageNet experiment.


**Limitations:**

1. There is no code provided to reproduce the results.
2. The authors do not describe limitations of their work.

---

> ### Author Rebuttal · Authors · 2023-08-07
>
> Thank you for your review; we are delighted that you liked the paper and found our results important and significant. In this paper, we focused on the theoretical aspect, but we agree that further empirical investigations is an exciting area for future research. We hope that the theoretical insights we have developed can help practitioners develop novel methods that perform better in the large scale regime.
>
> On computational cost, the FLOPs for an SGD update is 2N, N being the number of parameters in the network. The Optimistic Meta-Learning optimizer in the ImageNet experiment (Section 7.2) requires 7N FLOPs. As a reference, an optimizer like Adam requires 18N FLOPs per update.

---

> > ### Comment · Reviewer_nSnp · 2023-08-15
> > **Thanks to the authors for the rebuttal**
> >
> > I’ve read comments from all the other reviewers. Thank you for your rebuttal, and I appreciate that my concerns have been addressed.

---

### Official Review · Reviewer_JxCU · 2023-07-07

**Soundness:** 3 good
**Presentation:** 3 good
**Contribution:** 3 good
**Rating:** 6
**Confidence:** 3

**Summary:**

This work discovers the connection between gradient-based meta-learning and convex optimization of the meta parameters. From there, the authors observe that common gradient descent and its variants with momentum are special cases. To match the conventional $O(1 / T^2)$ convergence rate, the authors propose the optimistic version.

**Strengths:**

1. The observation that common accelerated methods can be viewed as meta-learning is interesting. The online convex optimization view presents a new perspective for showing the convergence of the accelerated methods, which could be itself interesting.
2. The proposed optimistic gradient-based meta-learning method is simple, but shows performance gain over vanilla baseline methods.

**Weaknesses:**

The major weakness of the analysis comes from the restrictive form of the meta-learner. This makes the analysis not applicable to all neural network based learn-to-optimize methods, while the meta-learning methods applied on learning rate or pre-conditioning matrix are well studied already in the literature.

**Questions:**

The figure 1 shows the result of top-1 accuracy of sgd and standard meta learning on resnet50 on imagenet. But to my knowledge SGD can achieve >75% top-1 accuracy on imagenet, can the author elaborate more on what codebase they use and the details of experimental setting for that experiment (e.g., how do you tune learning rate, what is the scheduling etc)?

**Limitations:**

The authors have adequately addressed the limitations.

---

> ### Author Rebuttal · Authors · 2023-08-07
>
> Thank you for your review, we are glad you found the connections we made interesting. We fully agree that the restrictions on the meta-learner are limiting. Unfortunately, this is an inherent limitation we face when using convex analysis, since neural networks are not convex. Hence, a non-convex interaction between the meta-learner and the learner cannot be analyzed exactly in this way. With that said, we would like to point out that our theoretical predictions do apply to any meta-learner, up to first order Taylor series error. For small learning rates, our predictions are accurate, and our ResNet experiment demonstrates that our theory has practical use.
>
> Thank you for raising your question on Figure 1. We use the Haiku example launch script, which is available on github. We are not allowed to provide external links in the rebuttal, but please see Appendix C in the supplementary materials for a url link. We only modified this script by changing the optimizer. The baseline we compare against is SGD with a fixed learning rate (a non-adaptive baseline). We tuned the learning via a sweep over the range $[1e-4, 3e-4, 1e-3, 3e-3, 1e-2, 3e-2, 1e-1, 3e-1]$ on the validation set. We report scores for the best performing one. We would like to stress that the point of this experiment is not to propose a new meta-learner that beats state of the art, but to empirically validate that our theoretical predictions (i.e. that an adaptive meta-learner can yield accelerated learning against a non-adaptive learner) hold in the deep learning setting.

---

> > ### Comment · Reviewer_JxCU · 2023-08-12
> > **Response to Authors' Rebuttal**
> >
> > Thanks for your rebuttal and my concerns are addressed. I hope that the implementations will be made publicly available for reproducibility.

---

### Official Review · Reviewer_HQ88 · 2023-07-10

**Soundness:** 4 excellent
**Presentation:** 3 good
**Contribution:** 3 good
**Rating:** 7
**Confidence:** 3

**Summary:**

The submission studies connections between recent advances in convex optimisation and heuristic meta-learning update rules. The provided framework contains standard methods such as heavy ball and Nesterov's momentum as special cases, while also containing rules that correspond to online meta-learning. Bootstrapped meta-gradients are also formalised within this framework. Some of the theoretical results are corroborated by empirical investigations.

**Strengths:**

* The submission is well-written and interesting. I can see myself referring back to this paper in future when developing new meta-learning or bi-level optimisation methods.
* Framework provides a useful tool for conceptualising and analysing meta-learning algorithms, as well as comparing them with existing (or novel) convex optimisation approaches.
* The analysis shows how one can obtain optimal $O(1/T^2)$ rates of convergence by using optimism. E.g., by predicting that the current gradient will be similar to the previous gradient.
* Experiments are provided that empirically corroborate the predictions of the theoretical analysis. In particular, the meta-learning variants of momentum and adagrad converge substantially faster than the conventional approaches.

**Weaknesses:**

* The analysis is limited to the convex case, which is understandable. However, it would be interesting to at least empirically investigate the non-convex case as well.

**Questions:**

N/A

**Limitations:**

There is no substantial discussion of the limitations of the work.

---

> ### Author Rebuttal · Authors · 2023-08-07
>
> Thank you for your review, we are glad you found the paper interesting and potentially helpful in your future work! We agree that studying the non-convex case empirically is an exciting area for future research. We take some initial steps in this direction with our ResNet experiment, in which the learner’s problem is non-convex (while the meta-learner’s problem is still convex). We hope that our theoretical insights can help practitioners develop new meta-learning algorithms that perform better at scale in non-convex settings.

---

### Decision · Program_Chairs · 2023-09-21

**Decision:**

Accept (poster)

**Comment:**

This work discovers the connection between gradient-based meta-learning and convex optimization of the meta parameters. From there, the authors observe that common gradient descent and its variants with momentum are special cases. To match the canonical optimal convergence rate, the authors propose an optimistic version to achieve it.

All reviewers believe this paper makes valid contributions. The AC agrees and recommends acceptance.